# Quantifying the impact of clinical coding in chronic kidney disease on risk of death and COVID-19 death

Stuart Stewart[1,2,3]*, Philip A. Kalra[1,3], Evangelos Kontopantelis[2], Tom Blakeman[2,4], George Tilston[5], Smeeta Sinha[1,3]

1 Donal O'Donoghue Renal Research Centre, Research & Innovation, Northern Care Alliance NHS Foundation Trust, Salford, England, United Kingdom, 2 Centre for Primary Care & Health Services Research, University of Manchester, Manchester, England, United Kingdom, 3 Division of Cardiovascular Sciences, University of Manchester, Manchester, England, United Kingdom, 4 National Institute for Health and Care Research (NIHR) Greater Manchester Patient Safety Research Collaboration (GM PSRC), Manchester, England, United Kingdom, 5 National Institute for Health and Care Research (NIHR) Manchester Biomedical Research Centre, University of Manchester, England, United Kingdom

* stuart.stewart@manchester.ac.uk

## Abstract

### Background

Patients with biochemical evidence of chronic kidney disease (CKD) without a diagnostic code (uncoded CKD) in primary care are at increased risk of death, acute kidney injury (AKI), and unplanned hospital care. Uncoded CKD is highly prevalent and there is no data to evaluate whether patients with uncoded CKD were at an increased risk of COVID-19 death. Aim: to assess whether patients with uncoded CKD stages 3–5 were at increased risk of death and COVID-19 deaths.

### Methods

Descriptive and inferential analyses to measure adjusted hazard of death, and COVID-19 death in patients with CKD stages 3–5 from 2.85 million primary care patients in Greater Manchester, England. Sensitivity analyses using propensity score matching and competing risk regression.

### Results

Coded CKD stages 3 and 4 (versus uncoded) were associated with significantly lower adjusted hazards of death (HR 0.81, CIs 0.77–0.86, p=<0.0001; HR 0.45, CIs 0.34–0.60, p=<0.0001, respectively), and COVID-19 death (HR 0.74, CIs 0.55–0.99, p=0.03; HR 0.55, CIs 0.30–0.99, p=0.045, respectively). Descriptive analyses were conducted for patients with CKD stage 5 due to low numbers of patients with uncoded CKD stage 5, precluding survival analyses.

**Data availability statement:** Researchers can apply to use the same GMCR data through an application to the GMCR board at the following address: https://healthinnovationmanchester.com/contact/.

**Funding:** Dr Stuart Stewart receives doctoral research funding from Kidney Research UK (AHP_001_202207405). The funder of the study had no role in article design, analysis, or publication writing.

**Competing interests:** I have read the journal's policy and the authors of this manuscript have the following competing interests: SSt receives doctoral research salary funding from Kidney Research UK. PK has received grant funding from CSL Vifor and Astellas, consulting fees from Astra Zeneca, Vifor, Unicyte, and UCB, honoraria from Astra Zeneca, Pfizer, Pharmacosmos, Medice, GSK, Bayer, and CSL Vifor. EK – None to declare. TB – no financial conflicts of interest; NHS England Think Kidneys Programme Board Member (2014–17); Royal College of General Practitioners' AKI Clinical Champion (2017–20); NHSE Renal Services Transformation Programme Post-AKI care Lead (2021–23); Specialist Committee Member for NICE AKI Quality Standard (QS76) (2022–23); Kidney Disease Improving Global Outcomes (KDIGO) AKI Guideline Work Group (2023-To date). GT receives salary funding by the National Institute for Health and Care Research (NIHR) Manchester Biomedical Research Centre (BRC) (NIHR 203308). SSi has received speaker fees from AstraZeneca, Bayer, Chiesi, Sanofi-Genzyme, Novartis, CSL Vifor, GSK, Menarini, Medscape, and Boehringer-Ingelheim; and consulting fees from AstraZeneca, Novo Nordisk, Amicus, Stada, Santhera, Novartis, Bayer, Sanofi-Genzyme, Vifor Pharma, GSK, Inozyme Pharma, Sobi, Inozyme Pharma Inc., Pursepring, and Boehringer-Ingelheim, and grant funding received from JnJ, AstraZeneca, CSL Vifor, Sanofi-Genzyme, and Novartis. This does not alter our adherence to PLOS ONE policies on sharing data and materials. There are no patents, products in development or marketed products associated with this research to declare.

## Conclusion

Our retrospective cohort study suggests that clinical coding is a digital intervention associated with a lower adjusted hazard of death and COVID-19 death in patients with CKD stages 3 and 4, and should be considered a key element in the organisation and delivery of care for people with CKD.

## Introduction

Chronic kidney disease (CKD) is highly prevalent in the UK [1], Europe and globally [2], predicted to be the fifth leading cause of life years lost by 2040 [1,2]. Primary care health services detect, diagnose and manage most CKD [3,4] across modern global health systems. Patients who are eventually diagnosed with CKD in primary care first undergo testing, followed by diagnosis and coding of that diagnosis in an electronic health record (EHR).

Clinical coding is essential for modern digital health records, maintaining accurate disease registers for clinicians and researchers [5,6]. By typing a standardised diagnostic code (e.g., SNOMED-CT [7], ICD-11 [8]) clinicians and patients benefit from automated monitoring, clinical target and vaccination reminders, and prescribing and cross-disease management alerts [4]. Clinical coding helps to operationalise complex evidence-based guidelines into actionable suggestions at key points of clinical decision-making. CKD coding is therefore associated with higher quality of care [9], reduced AKI risk, and hospitalisation [3,10,11]. Moreover, patients with uncoded CKD stage 3 (estimated glomerular filtration rate (eGFR) 42–43 mL/min/1.73m$^2$ rate ratio 1.87; CIs 1.63–2.16), stage 4 (eGFR 28–29 mL/min/1.73m$^2$ rate ratio 3.67; CIs 2.95–4.56) and stage 5 (eGFR 0–14 mL/min/1.73m$^2$ rate ratio 6.13, CIs 3.96–9.49) are at increased risk of death [10]. Despite these benefits, uncoded CKD is common in England [9] due to system, practitioner and patient level barriers, including funding constraints, limited clinician recognition or knowledge, and concerns regarding over medicalisation of ageing [4]. As such, identifying patients with uncoded CKD in primary care is a priority with national cardiovascular auditing tools (CVD PREVENT [12]) in England allowing quantification of uncoded CKD across practices and regions.

Identifying patients with CKD in primary care was essential during the COVID-19 pandemic due to increased COVID-19 mortality risk, informing tailored shielding advice [13] and COVID-19 vaccination priority [14]. Clinical coding was invaluable for GP practices and NHS England to identify clinically vulnerable patients [13,15,16]. Given an increased risk of death with uncoded CKD [3], it is hypothesised that uncoded CKD may also be associated with increased risk of COVID-19 death, however this is unexplored [4]. Determining this association is important for CKD care, population-level interventions and future pandemic preparedness.

## Methods

### Study design and participants

A retrospective observational cohort study using primary care EHR data from the Greater Manchester Care Record (GMCR; ref GMCR-RQ041) from 1st March

2018–1st August 2023. The GMCR pools EHR data for 2.85 million citizens across 433 general practices (99.7% of all practices) across Greater Manchester, England [17].

All data were de-identified at source and were extracted according to eligibility criteria. Inclusion criteria: adults (18+ years of age) with CKD stages 3−5 at study start, according to the National Institute of health and Care Excellence (NICE) guidelines using KDIGO criteria [18]. This included all patients with a diagnostic code for CKD stages 3−5 (coded CKD), and patients with biochemical evidence of CKD without a diagnostic code (uncoded CKD) – 2 x eGFR < 60 mL/min/1.73m$^2$, at least 90 days apart. Exclusion criteria: patients without a measured creatinine/eGFR value during the study period; patients entering the study period after 2019; patients with uncoded CKD at study start that were diagnosed after 2019; patients with codes for kidney transplant, dialysis, dementia or palliative care at study start and dementia codes during the study; and patients with CKD stages 1 and 2, and missing eGFRs in 2018 (Fig 1).

The aims of this research, codesigned with a CKD patient involvement group, were to quantify the impact of clinical coding on risk of death and COVID-19 death in patients with CKD stages 3–5, by CKD stage, in primary care in Greater Manchester, England.

### Ethical approval

Ethical approval for use of GMCR data was defined in the national Control of Patient Information (COPI) notice [19] allowing health record data to be used in COVID-19 related research.

### Procedures

Primary outcomes included all-cause mortality and COVID-19 death (within 28 days of a positive COVID-19 test within a patient's EHR).

Analyses were conducted by CKD stage (3, 4 and 5) at study start (2018) using eGFR, recalculated using CKD-EPI 2021 [20], to compare the effect of coding status (coded/uncoded) on outcomes within each CKD stage. For patients with multiple eGFRs within a year, the latest eGFR was chosen. To avoid including patients with AKI, all creatinine results with a corresponding AKI clinical code were excluded.

Predictor variables included sex, age group (18−39; 40−59; 60−74; 75−89; 90+), ethnicity (White or White British; Asian or Asian British; Black or Black British; Mixed; Other ethnic groups), body mass index (BMI), socioeconomic status as measured using the Index of Multiple Deprivation (IMD) deciles (a measure of geographical area level deprivation at a low geographical level of approximately 1600 people), and clinical diagnostic codes at study start (diabetes, hypertension, coronary heart disease (CHD), heart failure, peripheral arterial disease (PAD), stroke, transient ischaemic attack (TIA), gout, myeloma, non-alcoholic fatty liver disease (NAFLD), systemic lupus erythematosus (SLE), osteoporosis, glomerulonephritis, vasculitis, autosomal dominant polycystic kidney disease (ADPKD), kidney stones, AKI, depression, schizophrenia, bipolar disorder, eating disorder, self-harm and suicidal ideation), CKD coding status, and COVID-19 vaccination status (Supporting information 1).

### Statistical analyses

In primary analyses, adjusted Cox proportional hazard (PH) models quantified the association between clinical coding in CKD stages 3 and 4 on, a) all-cause mortality and b) COVID-19 mortality, controlling for several covariates. For hazard of all-cause mortality, all patients were included. For hazard of COVID-19 mortality, in order to isolate the impact of coding on the risk of COVID-19 death, non-COVID deaths after either a) the first suspected COVID-19 death in the UK (01/03/2020) or b) the date of first vaccine availability in the UK (8/12/2020), were excluded. Time-to-event (TTE) for all-cause mortality was calculated as death date or censoring at study end (31/08/2023) – start date (01/03/2018), in days. For COVID-19 mortality, TTE was death date or censoring at study end (31/08/2023) – (01/03/2020 or 8/12/2020). Predictors violating the PH assumption (ethnicity, age group, BMI), were stratified in Cox PH models to allow for separate baseline hazard

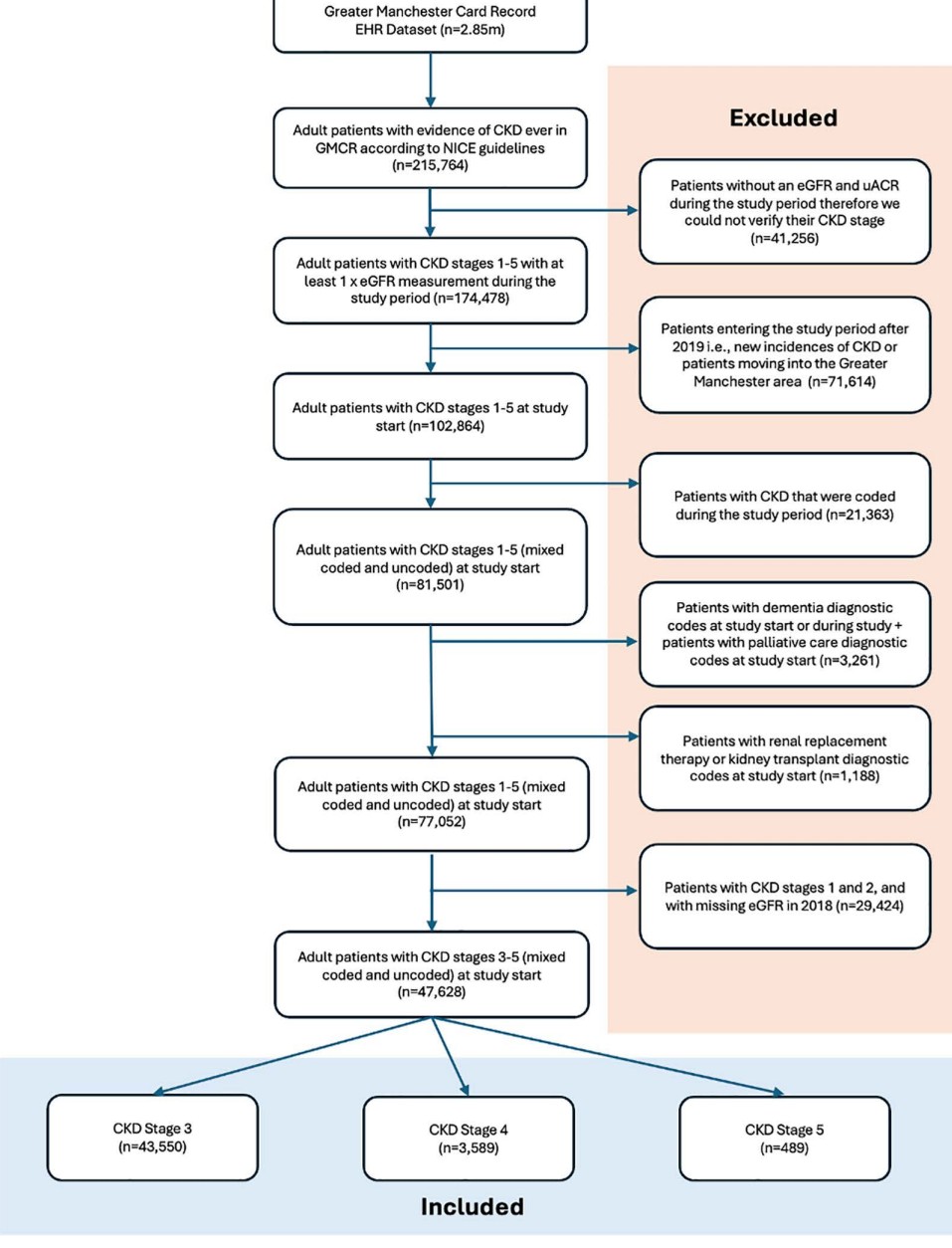

**Fig 1. Study participant flow diagram showing inclusion and exclusion of patients with CKD from the Greater Manchester Care Record dataset.**

functions. COVID-19 analyses for stage 4 CKD included all patients in 2018 and 2019 to increase number of patients. Survival analyses were conducted by CKD stage to allow for coding status to be compared within each group, in line with existing research in this area [3,10,21].

Sensitivity analyses used Fine-Gray competing risk regression (CRR) to measure the impact of coding on hazard of COVID-19 mortality with competing non-COVID mortality risk, and addressed covariate imbalance through propensity score matching (PSM), between patients with coded and uncoded CKD, with a calliper of 0.1 and using covariates (sex,

age group, BMI, IMD decile, ethnicity, and comorbidities) to predict group membership. Group differences are reported using standardised mean differences (SMDs).

Descriptive analyses were conducted for patients with CKD stage 5 due to low numbers of uncoded CKD stage 5, precluding survival analyses. Continuous variables were described using median and interquartile range (IQR). Categorical variables were described using frequency and percentage. Complete cases were analysed due to computational limitations for multiple imputation of missing eGFR and uACR data. Participants with missing BMI and IMD data were dropped; those with missing ethnicity data were categorised into a 'missing ethnicity' group. To protect patient confidentiality, cell counts less than five were suppressed and reported as '<5'.

All analyses were undertaken using R and RStudio (version 4.1.0) (R Foundation for Statistical Computing, Vienna, Austria). Statistical significance was pre-specified at 0.05.

## Results

There were 47,628 patients with CKD stage 3 (n = 43,550; 91.4%), stage 4 (n = 3,589; 7.5%), and stage 5 (n = 489; 1.1%). Male prevalence increased with CKD severity (53.3% stage 3; 68.5% stage 5) (Table 1). Median age was highest in CKD stage 4 at 78 (IQR 70–85) years, and lowest in stage 5 (68 years, IQR 56–79).

Most patients were of overweight or obese BMI. White or White British ethnicity was most common (78.8% in stage 3, 73.4% in stage 4, and 61.8% in stage 5), with Asian and Black ethnicities increasing as CKD stage increased.

Hypertension (69.7% stage 3; 83.8% stage 5) and diabetes (30.4% stage 3; 44.2% stage 5) were the most prevalent comorbidities. AKI prevalence increased from 16.6% in stage 3 to 30.5% in stage 5. Depression was the most prevalent mental health condition (30.4% stage 3; 26% stage 4; 29.4% stage 5). Across all stages, over 45% of patients lived in the 3 most deprived IMD deciles. The degree of albuminuria increased with CKD stage (Table 1). Prevalence of covariates by coding status for CKD stages 3 and 4 are presented in Supporting information 2.

### Descriptive analyses for CKD stage 5

There were 489 patients with CKD Stage 5; 68.5% male; 34.6% were overweight and 28.6% were of obese BMI. Median age was 68 years (IQR 56–79). Most were White (61.8%), followed by Asian (17.2%), other ethnic groups (10.6%), Black (7.0%), and mixed (1.2%) ethnicities. Approximately 60% of patients lived in the 3 most deprived IMD deciles. Median eGFR was 11 mL/min/1.73m$^2$ (IQR 8–13) and median uACR was 39.3 mg/mmol (IQR 9.0–97.7). Coded CKD stage 5 at study start was most common (98.8%).

Hypertension and diabetes were prevalent in 83.8% and 44.2% of patients, respectively. Other prevalent comorbidities were AKI (30.5%), depression (29.4%), CHD (22.5%), ADPKD (19.4%), gout (24.3%), CHD (22.5%), heart failure (13.1%), glomerulonephritis (5.7%), stroke (8.4%), PAD (8.0%), TIA (4.9%), osteoporosis (4.1%), self-harm and suicidal ideation (3.3%), eating disorder (3.3%), kidney stones (3.3%), NAFLD (1.2%), vasculitis (1.2%), schizophrenia (<1.0%), SLE (<1.0%), myeloma (<1.0%), and bipolar disorder (<1.0%). The crude cumulative mortality rate was 37.2% with a COVID-19 crude cumulative mortality rate of 1.5% (of 472 patients alive as of 01/03/2020).

### Impact of coding on risk of death

**Stage 3.** Among 34,863 coded and 8,080 uncoded patients with CKD stage 3, crude cumulative mortality rates were 20.8% versus 19.9%, respectively.

Coded CKD stage 3 was associated with a significantly lower adjusted hazard of death (HR 0.81, CIs 0.77–0.86, p=<0.0001) than uncoded CKD stage 3. Other significant predictors included male sex (HR 1.10), increasing age, low BMI (HR 1.75), overweight BMI (HR 0.71), obese BMI (HR 0.70), IMD decile (HR 0.95), diabetes (HR 1.37), hypertension (HR 1.13), gout (HR 1.09), osteoporosis (HR 1.19), CHD (HR 1.24), heart failure (HR 1.64), PAD (HR 1.48), stroke (HR 1.37), TIA (HR 1.14), AKI (HR 1.08), depression (HR 1.11), schizophrenia (HR 1.54), and eating disorder (HR 1.19) (Table 2, Fig 2).

**Table 1. Clinical and demographic summary of cohort by CKD stage.**

| Variables | CKD stage 3 | CKD stage 4 | CKD stage 5 |
|---|---|---|---|
| | N (%) | N (%) | N (%) |
| Total patients | 43550 (91.4) | 3589 (7.5) | 489 (1.1) |
| **Sex** | | | |
| Male | 23232 (53.3) | 2446 (68.2) | 335 (68.5) |
| **Age** | | | |
| Age group 18–39 | 298 (0.7) | 53 (1.3) | 28 (5.7) |
| Age group 40–59 | 4072 (9.4) | 327 (9.1) | 121 (24.7) |
| Age group 60–74 | 15783 (36.2) | 948 (26.4) | 178 (36.4) |
| Age group 75–89 | 21086 (48.4) | 1931 (53.8) | 155 (31.7) |
| Age group 90+ | 2311 (5.3) | 330 (9.2) | 7 (1.4) |
| Median (IQR) years | 75 (68-82) | 78 (70-85) | 68 (56-79) |
| **BMI group** | | | |
| Low BMI < 18.5 | 569 (1.3) | 58 (1.6) | 7 (1.4) |
| Normal BMI 18.5–24.9 | 9838 (22.6) | 868 (24.2) | 128 (26.2) |
| Overweight BMI 25–29.9 | 16746 (38.5) | 1328 (37.0) | 169 (34.6) |
| Obese BMI 30–39.9 | 14003 (32.2) | 1090 (30.4) | 140 (28.6) |
| Severely obese BMI >=40 | 1794 (4.1) | 183 (5.1) | 28 (5.7) |
| Missing | 600 (1.4) | 62 (1.7) | 17 (3.5) |
| **Ethnicity** | | | |
| White or White British | 34318 (78.8) | 2635 (73.4) | 302 (61.8) |
| Asian or Asian British | 2513 (5.8) | 310 (8.6) | 84 (17.2) |
| Black or Black British | 1169 (2.7) | 96 (2.7) | 34 (7.0) |
| Mixed | 280 (0.6) | 19 (0.5) | 6 (1.2) |
| Other ethnic groups | 4370 (10.0) | 438 (12.2) | 52 (10.6) |
| Missing | 900 (2.1) | 91 (2.5) | 11 (2.2) |
| **IMD deciles** | | | |
| 1 (most deprived) | 8824 (20.3) | 823 (22.9) | 158 (32.3) |
| 2 | 6079 (14.0) | 516 (14.4) | 73 (14.9) |
| 3 | 4995 (11.5) | 368 (10.3) | 65 (13.3) |
| 4 | 3504 (8.0) | 307 (8.6) | 30 (6.1) |
| 5 | 3512 (8.1) | 262 (7.3) | 26 (5.3) |
| 6 | 2674 (6.1) | 199 (5.5) | 36 (7.4) |
| 7 | 3553 (8.2) | 309 (8.6) | 31 (6.3) |
| 8 | 4186 (9.6) | 346 (9.6) | 26 (5.3) |
| 9 | 3502 (8.0) | 278 (7.7) | 26 (5.3) |
| 10 (least deprived) | 2714 (6.2) | 180 (8.0) | 18 (3.7) |
| Missing | 7 (<0.1) | <5 (<0.1) | <5 (<0.1) |
| **Diagnoses at study start** | | | |
| Diabetes | 13228 (30.4) | 1682 (46.9) | 216 (44.2) |
| Hypertension | 30363 (69.7) | 2901 (80.8) | 410 (83.8) |
| SLE | 130 (0.3) | 15 (0.4) | <5 (<1.0) |
| Gout | 5886 (13.5) | 910 (25.4) | 119 (24.3) |
| NAFLD | 666 (1.6) | 52 (1.4) | 6 (1.2) |
| Myeloma | 96 (0.2) | 18 (0.5) | <5 (<1.0) |
| Osteoporosis | 3587 (8.2) | 251 (7.0) | 20 (4.1) |
| CHD | 9670 (22.2) | 1070 (29.8) | 110 (22.5) |

*(Continued)*

**Table 1.** (Continued)

| Variables | CKD stage 3 | CKD stage 4 | CKD stage 5 |
|---|---|---|---|
| Heart failure | 3684 (8.5) | 578 (16.1) | 64 (13.1) |
| PAD | 2052 (4.7) | 291 (8.1) | 39 (8.0) |
| Stroke | 3285 (7.5) | 353 (9.8) | 41 (8.4) |
| TIA | 2456 (5.6) | 243 (6.8) | 24 (4.9) |
| ADPKD | 6.5 (1.4) | 247 (6.9) | 95 (19.4) |
| Glomerulonephritis | 177 (0.4) | 91 (2.5) | 28 (5.7) |
| Kidney stones | 1035 (2.4) | 127 (3.5) | 16 (3.3) |
| Vasculitis | 295 (0.7) | 42 (1.2) | 6 (1.2) |
| Acute kidney injury | 7229 (16.6) | 824 (23.0) | 149 (30.5) |
| Depression | 13223 (30.4) | 938 (26.0) | 144 (29.4) |
| Schizophrenia | 851 (2.0) | 71 (2.0) | <5 (<1.0) |
| Bipolar disorder | 370 (0.8) | 34 (0.9) | <5 (<1.0) |
| Eating disorder | 684 (1.6) | 69 (1.9) | 16 (3.3) |
| Self-harm and suicidal ideation | 1267 (2.9) | 87 (2.4) | 16 (3.3) |
| Coding status | | | |
| Coded at study start | 35307 (81.1) | 3512 (97.9) | 483 (98.8) |
| COVID-19 vaccination | | | |
| Vaccinated | 41354 (95.0) | 3322 (92.6) | 435 (89.0) |
| Measurements | | | |
| eGFR (median(IQR)) | 48 (41-53) | 25 (22-28) | 11 (8-13) |
| uACR (median(IQR)) | 1.7 (0.8-5.0) | 6.31 (1.7-23.7) | 39.25 (9.0-97.7) |

**Key:** BMI, body mass index; SLE, systemic lupus erythematosus; NAFLD, non-alcohol fatty liver disease; CHD, coronary heart disease; PAD, peripheral arterial disease; TIA, transient ischaemic attack; ADPKD, autosomal dominant polycystic kidney disease; eGFR, estimated glomerular filtration rate; uACR, urine albumin creatinine ratio.

**Stage 4.** Among 3,451 coded and 75 uncoded patients with CKD stage 4, crude cumulative mortality rates were 38.2% and 66.6%, respectively.

Coded CKD stage 4 was associated with a significantly lower adjusted hazard of death (HR 0.45, CIs 0.34–0.60, p=<0.0001) than uncoded CKD stage 4. Other significant predictors of death include male sex (HR 0.89), increasing age, low BMI (HR 1.66), overweight BMI (HR 0.74), obese BMI (HR 0.75), IMD decile (HR 0.96), diabetes (HR 1.19), CHD (HR 1.27), heart failure (HR 1.72), PAD (HR 1.57), stroke (HR 1.40), and eating disorder (HR 1.59) (Table 2, Fig 3).

## Impact of coding on risk of COVID-19 death

**Stage 3.** There were 27,871 coded and 6,537 uncoded patients with CKD stage 3; with a crude cumulative mortality rate of 0.9% for both groups.

Coded CKD stage 3 was associated with a significantly lower adjusted hazard of a COVID-19 death (HR 0.74, CIs 0.55–0.99, p=0.03) than uncoded CKD stage 3. Other significant predictors included age group, low BMI (HR 2.62), IMD decile (HR 0.92), diabetes (HR 1.82), gout (HR 1.38), CHD (HR 1.32), heart failure (HR 1.53), PAD (HR 1.75), stroke (HR 1.88), depression (HR 1.32) and schizophrenia (HR 2.10) (Table 3, Figs 4 and 5).

**Stage 4.** There were 1,887 coded and 355 uncoded patients with CKD stage 4; with 2.8%, and 3.9% crude cumulative mortality rates, respectively.

**Table 2. Adjusted Cox proportional hazards model of impact of coding and other predictors on hazard of all-cause mortality in patients with coded and uncoded CKD stages 3 and 4.**

| | Adjusted hazard of all-cause mortality | | | | | |
| | CKD Stage 3 | | | CKD Stage 4 | | |
| Variable | Hazard ratio | 95% CIs | p-value | Hazard ratio | 95% CIs | p-value |
|---|---|---|---|---|---|---|
| Coded CKD | 0.81 | 0.77-0.86 | **<0.0001** | 0.45 | 0.34-0.60 | **<0.0001** |
| Male Sex | 1.10 | 1.04-1.15 | **0.0002** | 0.89 | 0.79-1.00 | **0.05** |
| Age group 18–39 | 0.21 | 0.09-0.50 | **0.0004** | 0.17 | 0.04-0.70 | **0.01** |
| Age group 40–59 | 0.41 | 0.34-0.48 | **0.0001** | 0.43 | 0.30-0.63 | **<0.0001** |
| Age group 75–89 | 2.77 | 2.62-2.94 | **<0.0001** | 2.03 | 1.75-2.36 | **<0.0001** |
| Age group 90+ | 7.73 | 7.15-8.35 | **<0.0001** | 4.67 | 1.75-2.36 | **<0.0001** |
| Low BMI<18.5 | 1.75 | 1.55-1.98 | **<0.0001** | 1.66 | 1.18-2.33 | **0.004** |
| Overweight BMI 25–29.9 | 0.71 | 0.68-0.75 | **<0.0001** | 0.74 | 0.65-0.87 | **<0.0001** |
| Obese BMI 30–39.9 | 0.70 | 0.66-0.75 | **<0.0001** | 0.75 | 0.65-0.87 | **0.0002** |
| Severely obese BMI >=40 | 1.04 | 0.93-1.17 | 0.45 | 0.98 | 0.75-1.29 | 0.88 |
| IMD decile | 0.95 | 0.94-0.96 | **<0.0001** | 0.96 | 0.94-0.98 | **<0.0001** |
| Diabetes | 1.37 | 1.31-1.43 | **<0.0001** | 1.19 | 1.06-1.33 | **0.003** |
| Hypertension | 1.13 | 1.07-1.18 | **<0.0001** | – | – | – |
| Gout | 1.09 | 1.03-1.16 | **0.003** | – | – | – |
| Osteoporosis | 1.19 | 1.11-1.27 | **<0.0001** | – | – | – |
| CHD | 1.24 | 1.18-1.30 | **<0.0001** | 1.27 | 1.13-1.43 | **<0.0001** |
| Heart Failure | 1.64 | 1.55-1.74 | **<0.0001** | 1.72 | 1.51-1.96 | **<0.0001** |
| PAD | 1.48 | 1.37-1.60 | **<0.0001** | 1.57 | 1.33-1.85 | **<0.0001** |
| Stroke | 1.37 | 1.29-1.47 | **<0.0001** | 1.40 | 1.21-1.63 | **<0.0001** |
| TIA | 1.14 | 1.06-1.23 | **0.0006** | – | – | – |
| AKI | 1.08 | 1.02-1.14 | **0.005** | – | – | – |
| Depression | 1.11 | 1.06-1.16 | **<0.0001** | – | – | – |
| Schizophrenia | 1.54 | 1.33-1.77 | **<0.0001** | – | – | – |
| Eating disorder | 1.19 | 1.04-1.36 | **0.01** | 1.59 | 1.13-2.23 | **0.008** |
| Reference categories | Coding status: uncoded at study start; Sex: Female; Age group: 60–74; BMI: 18.5–24.9 (healthy); IMD: 1 decile increase; Comorbidities: absence of diagnosis at study start. | | | | | |
| Stratified variables | Ethnicity group as violated PH assumption | | | | | |
| Statistically significant | P-values in bold | | | | | |
| Statistically non-significant | Variables not in the model or cells including '-' | | | | | |

Coded CKD stage 4 was associated with a borderline significantly lower adjusted hazard of death (HR 0.55, CIs 0.30–0.99, p=0.045) than uncoded CKD stage 4. Other significant predictors included age group 75–89 years (HR 1.86), age group 90+years (HR 4.64), heart failure (HR 2.36), and stroke (HR 2.18) (Table 3, Figs 6 and 7).

## Impact of coding and COVID-19 vaccination on risk of COVID-19 death

**Stage 3.** Excluding patients that died before the first available COVID-19 vaccine, there were 34,354 patients with CKD stage 3; 33,337 (97.0%) were vaccinated and 1017 (3.0%) were unvaccinated against COVID-19; with 0.7% and 4.2% mortality rates, respectively.

Coded CKD stage 3 remained a significant predictor of a lower adjusted hazard of COVID-19 mortality (HR 0.64, CIs 0.47–0.87, p=0.004) after accounting for COVID-19 vaccination status. Vaccinated patients had a significantly lower

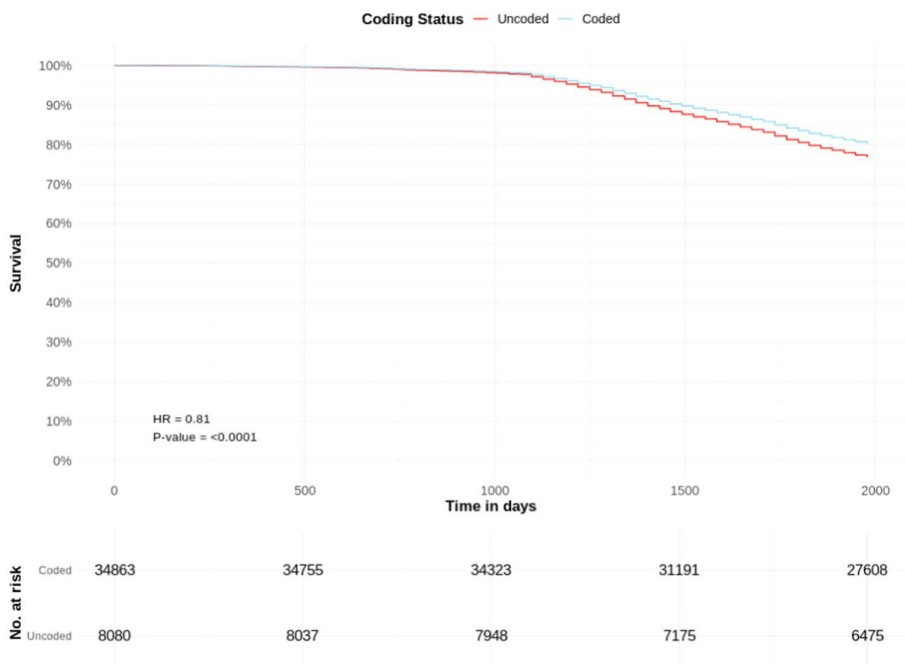

**Fig 2. All-cause mortality risk in patients with coded and uncoded CKD Stage 3: Survival curves from adjusted Cox proportional hazards model (Table 2).** T0 = March 1st, 2018.

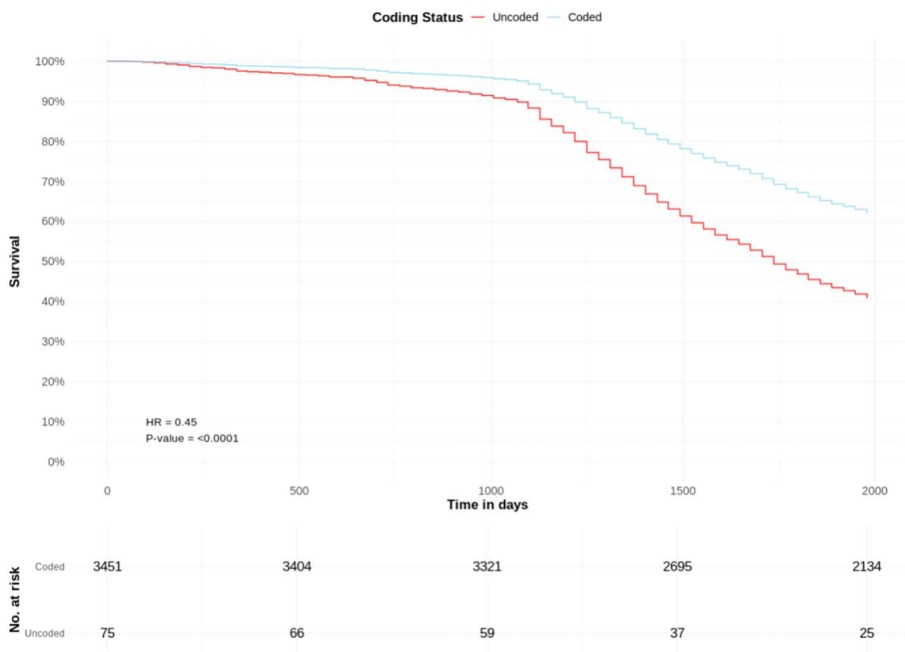

**Fig 3. All-cause mortality risk in patients with coded and uncoded CKD Stage 4: Survival curves from adjusted Cox proportional hazards model (Table 2).** T0 = March 1st, 2018.

**Table 3. Adjusted Cox proportional hazards model of impact of coding and other predictors on hazard of COVID-19 mortality in patients with coded and uncoded CKD stages 3 and 4.**

| | Adjusted hazard of COVID-19 death | | | | | |
| | Stage 3 | | | Stage 4 | | |
| Variable | Hazard ratio | 95% CIs | p-value | Hazard ratio | 95% CIs | p-value |
|---|---|---|---|---|---|---|
| **Coded CKD** | 0.74 | 0.55-0.99 | **0.03** | 0.55 | 0.30-0.99 | **0.045** |
| **Male Sex** | 1.09 | 0.86-1.40 | 0.47 | 0.83 | 0.48-1.43 | 0.50 |
| **Age group 18–39** | 1.01 | 0.14-7.34 | 0.99 | 1.05 | 0.13-8.38 | 0.96 |
| **Age group 40–59** | 0.23 | 0.08-0.64 | **0.004** | 0.55 | 0.16-1.96 | 0.36 |
| **Age group 75–89** | 3.13 | 2.35-4.16 | **<0.0001** | 1.86 | 1.00-3.45 | **0.05** |
| **Age group 90+** | 13.58 | 9.06-20.35 | **<0.0001** | 4.64 | 1.81-11.93 | **0.001** |
| **Low BMI<18.5** | 2.62 | 1.34-5.10 | **0.004** | 3.02 | 0.37-24.40 | 0.30 |
| **Overweight BMI 25–29.9** | 0.85 | 0.64-1.13 | 0.27 | 1.92 | 0.91-4.06 | 0.09 |
| **Obese BMI 30–39.9** | 0.94 | 0.69-1.27 | 0.69 | 1.91 | 0.88-4.14 | 0.10 |
| **Severely obese BMI>=40** | 1.19 | 0.69-1.27 | 0.57 | 1.40 | 0.29-6.60 | 0.68 |
| **IMD decile** | 0.92 | 0.88-0.96 | **<0.0001** | – | – | – |
| **Diabetes** | 1.82 | 1.45-2.29 | **<0.0001** | – | – | – |
| **Gout** | 1.38 | 1.03-1.86 | **0.03** | – | – | – |
| **CHD** | 1.32 | 1.03-1.69 | **0.03** | – | – | – |
| **Heart Failure** | 1.53 | 1.10-2.13 | **0.01** | 2.36 | 1.35-4.13 | **0.002** |
| **PAD** | 1.75 | 1.18-2.58 | **0.005** | – | – | – |
| **Stroke** | 1.88 | 1.38-2.57 | **<0.0001** | 2.18 | 1.07-4.47 | **0.03** |
| **Depression** | 1.32 | 1.04-1.68 | **0.02** | – | – | – |
| **Schizophrenia** | 2.10 | 1.11-3.97 | **0.02** | – | – | – |
| **Reference categories** | Coding status: uncoded at study start; Sex: Female; Age group: 60–74; BMI: 18.5–24.9 (healthy); IMD: 1 decile increase; Comorbidities: absence of diagnosis at study start. | | | | | |
| **Stratified variables** | Ethnicity group violated PH assumption. | | | | | |
| **Statistically significant** | P-values in bold. | | | | | |
| **Statistically non-significant** | Variables not in the model or cells including - | | | | | |

adjusted hazard of a COVID-19 death (HR 0.13, CIs 0.09–0.19, p=<0.0001) compared to unvaccinated patients. Other significant predictors included IMD (HR 0.94) and several comorbidities (Table 4, Fig 8).

**Stage 4.** Excluding patients that died before the first available COVID-19 vaccination, there were 2227 patients with stage 4 CKD; 2,178 (97.8%) were vaccinated and 49 (2.2%) were unvaccinated against COVID-19; with 10.5% and 8.2% mortality rates, respectively.

Coded CKD stage 4 remained a significant predictor of a lower adjusted hazard of COVID-19 mortality (HR 0.45, CIs 0.23–0.91, p=0.03) after accounting for COVID-19 vaccination status. Patients vaccinated against COVID-19 had a significantly lower adjusted hazard of a COVID-19 death (HR 0.17, CIs 0.06–0.50, p=0.001) compared to unvaccinated patients. Here, PAD is the only significant predictor (HR 3.30) (Table 4, Fig 9).

## Sensitivity analyses

In Fig 10, primary (model 1) and sensitivity analyses using PSM (models 2 and 3) show coded CKD stages 3 and 4 were associated with significantly lower adjusted hazard of death than uncoded CKD. Primary analyses (model 4) show a significantly lower adjusted hazard of a COVID-19 death for coded CKD stage 3, while CRR analyses with and without PSM (models 5–7), showed similar but non-significant results. Both primary (model 4) and CRR analyses (models 6 and

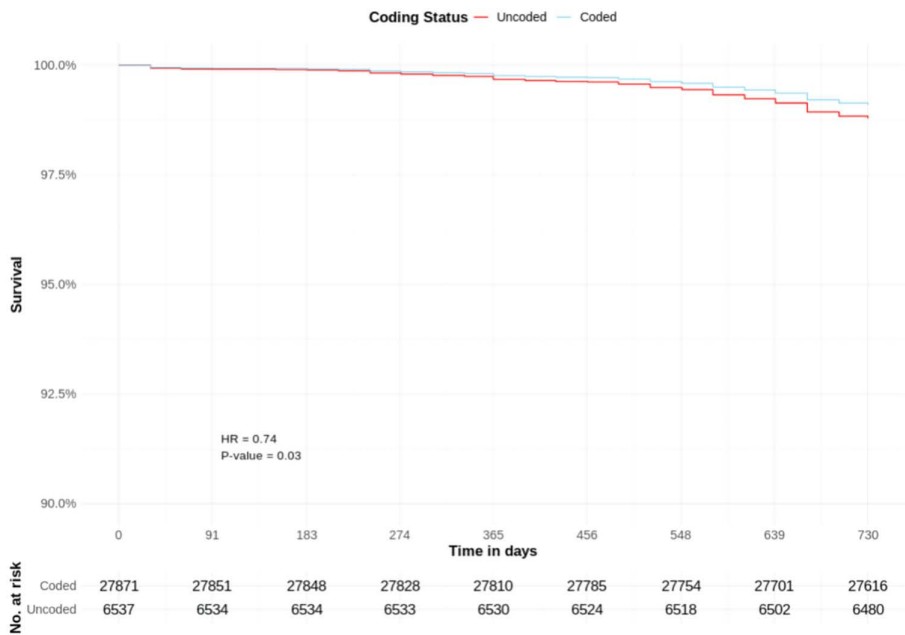

**Fig 4. COVID-19 mortality risk in patients with coded and uncoded CKD Stage 3: Survival curves from adjusted Cox proportional hazards model (Table 3) – adjusted y-axis 90%−100%.** T0 = March 1st, 2020.

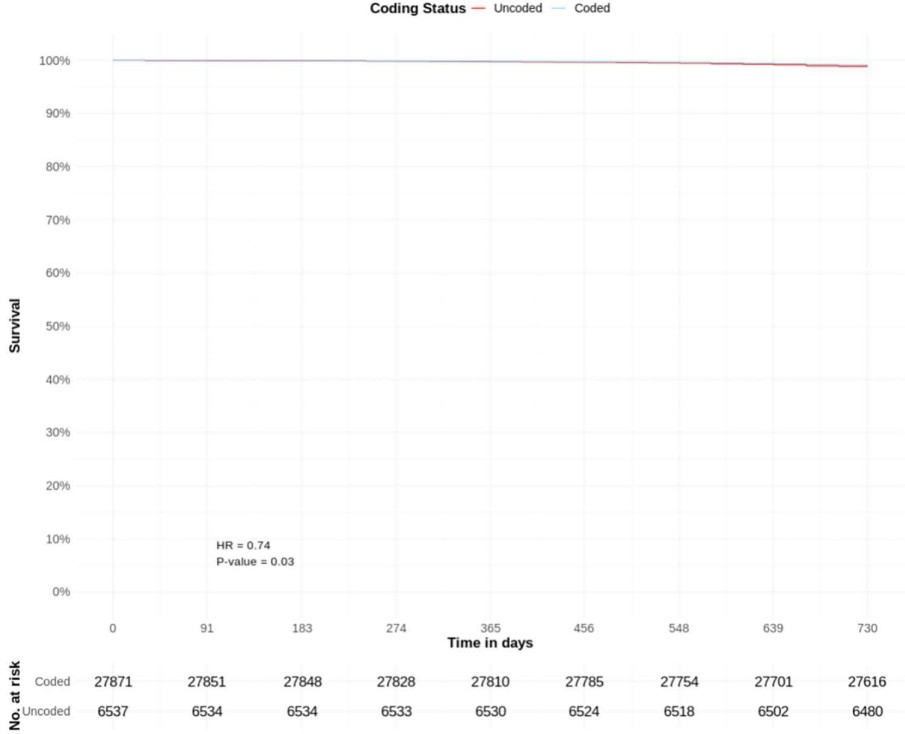

**Fig 5. COVID-19 mortality risk in patients with coded and uncoded CKD Stage 3: Survival curves from adjusted Cox proportional hazards model (Table 3) – full y-axis.** T0 = March 1st, 2020.

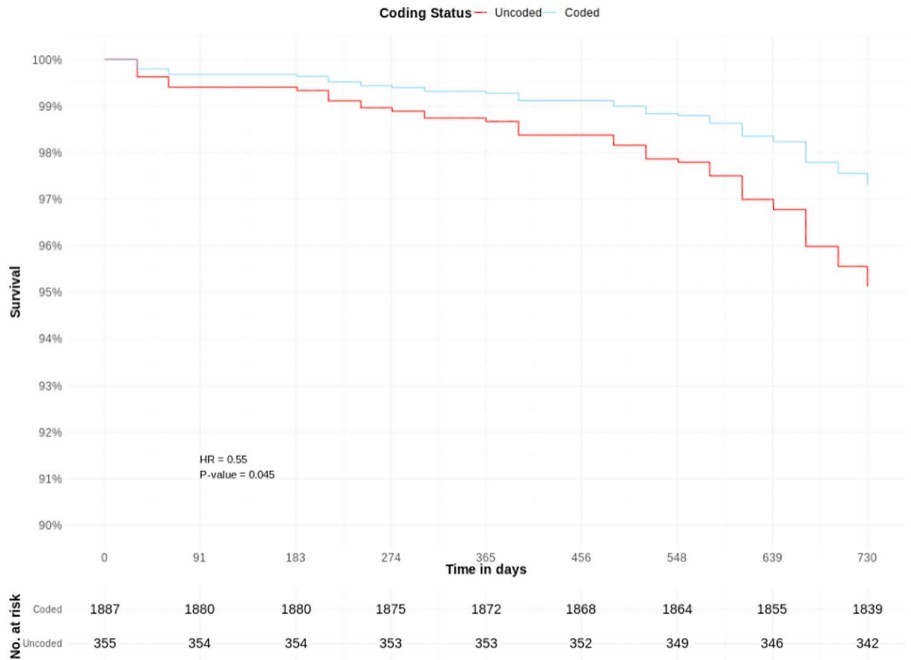

**Fig 6. COVID-19 mortality risk in patients with coded and uncoded CKD Stage 4: Survival curves from adjusted Cox proportional hazards model (Table 3) – adjusted y-axis 90%−100%.** T0 = March 1st, 2020.

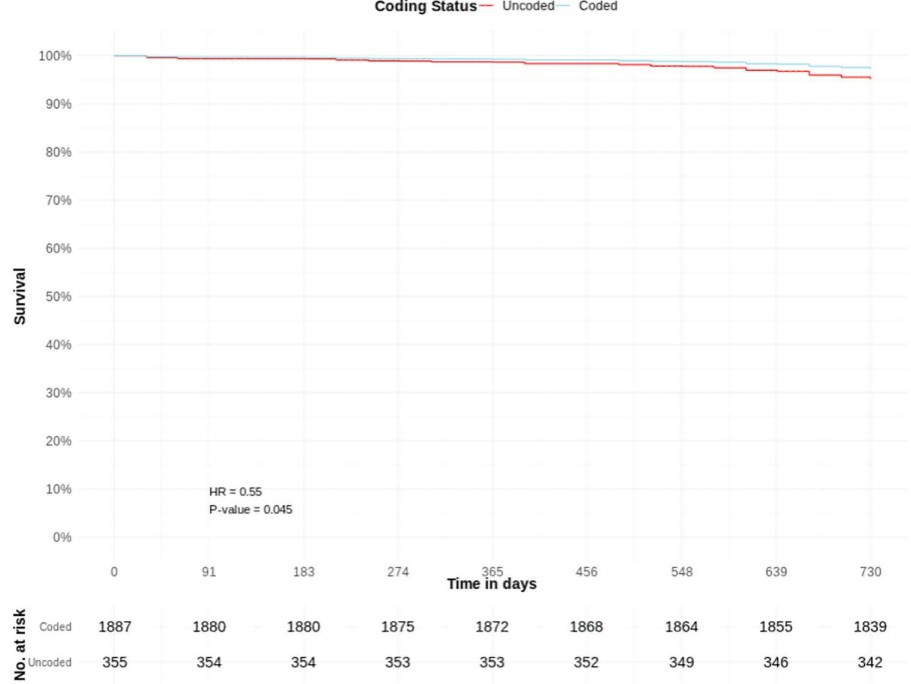

**Fig 7. COVID-19 mortality risk in patients with coded and uncoded CKD Stage 4: Survival curves from adjusted Cox proportional hazards model (Table 3) – full y-axis.** T0 = March 1st, 2020.

**Table 4. Adjusted Cox proportional hazards model of impact of coding, COVID-19 vaccination and other predictors on hazard of COVID-19 death in patients with CKD stages 3 and 4. T0 = December 8th, 2020.**

| | Hazard of COVID-19 mortality accounting for COVID-19 vaccinations | | | | | |
| | Stage 3 | | | Stage 4 | | |
| Variable | Hazard ratio | 95% CIs | p-value | Hazard ratio | 95% CIs | p-value |
|---|---|---|---|---|---|---|
| Coded CKD | 0.64 | 0.47-0.87 | **0.004** | 0.45 | 0.23-0.91 | **0.03** |
| COVID-19 vaccinated | 0.13 | 0.09-0.19 | **<0.0001** | 0.17 | 0.06-0.50 | **0.001** |
| Male Sex | 1.05 | 0.80-1.38 | 0.72 | 0.64 | 0.34-1.19 | 0.16 |
| Low BMI < 18.5 | 1.73 | 0.74-4.04 | 0.21 | – | – | – |
| Overweight BMI 25–29.9 | 0.86 | 0.63-1.18 | 0.34 | – | – | – |
| Obese BMI 30–39.9 | 1.05 | 0.76-1.45 | 0.78 | – | – | – |
| Severely obese BMI >=40 | 1.11 | 0.54-2.26 | 0.78 | – | – | – |
| IMD decile | 0.94 | 0.90-0.98 | **0.002** | – | – | – |
| Diabetes | 1.87 | 1.45-2.40 | **<0.0001** | – | – | – |
| Gout | 1.42 | 1.03-1.97 | **0.03** | – | – | – |
| CHD | 1.44 | 1.10-1.87 | **0.007** | – | – | – |
| PAD | 1.89 | 1.24-2.88 | **0.003** | 3.30 | 1.44-7.53 | **0.005** |
| Stroke | 1.61 | 1.12-2.33 | **0.01** | – | – | – |
| TIA | 1.50 | 1.01-2.22 | **0.04** | – | – | – |
| AKI | 1.44 | 1.08-1.93 | **0.01** | – | – | – |
| Depression | 1.31 | 1.01-1.70 | **0.04** | – | – | – |
| Bipolar disorder | 3.57 | 1.45-8.75 | **0.006** | – | – | – |
| Reference categories | Coding status: uncoded at study start; Sex: Female; Age group: 60–74; BMI: 18.5–24.9 (healthy); IMD: 1 decile increase; Comorbidities: absence of diagnosis at study starts. | | | | | |
| Stratified variables | Ethnicity and age group variables violated the PH assumption. | | | | | |
| Statistically significant | P-values in bold. | | | | | |
| Statistically non-significant | Variables not in the model or cells including '-' | | | | | |

7) showed a significantly lower adjusted hazard of COVID-19 death for coded CKD stage 4. In all models (8–10) including COVID-19 vaccinations (except CKD stage 3 model 9 PSM 1:1), coding remained associated with a significantly lower adjusted hazard of COVID-19 death. When balancing COVID-19 vaccination status across coded and uncoded cohorts in PSMs, vaccination status was not a significant predictor of COVID-19 death in CKD stage 4 (Table 14 in Supporting information 2), but it was significant for stage 3 (Table 6 in Supporting information 3).

## Discussion

To our knowledge, we present the first evidence that coded CKD stage 3 was associated with a significantly lower adjusted hazard of COVID-19 death (HR 0.74, CIs 0.55–0.99, p = 0.03) and coded CKD stage 4 was associated with a borderline significantly lower adjusted hazard (HR 0.55, CIs 0.30–0.99, p = 0.045), compared to uncoded CKD when non-COVID deaths are excluded. Sensitivity analyses examining competing risk indicated coded CKD stage 3 was not significantly associated with a lower adjusted hazard of COVID-19 death, indicating non-COVID-19 deaths were an important competing risk. We show that coded CKD stage 4 was associated with a significantly lower adjusted hazard of COVID-19 death whilst accounting for competing risks. Additionally, primary and sensitivity analyses using PSM (Fig 9: forest plot models 1−3) showed coded CKD stages 3 and 4 were associated with significantly lower adjusted hazards of all-cause mortality (stage 3 HR 0.81, CIs 0.77–0.86, p=<0.0001; stage 4 HR 0.45, CIs 0.34–0.60, p=<0.0001).

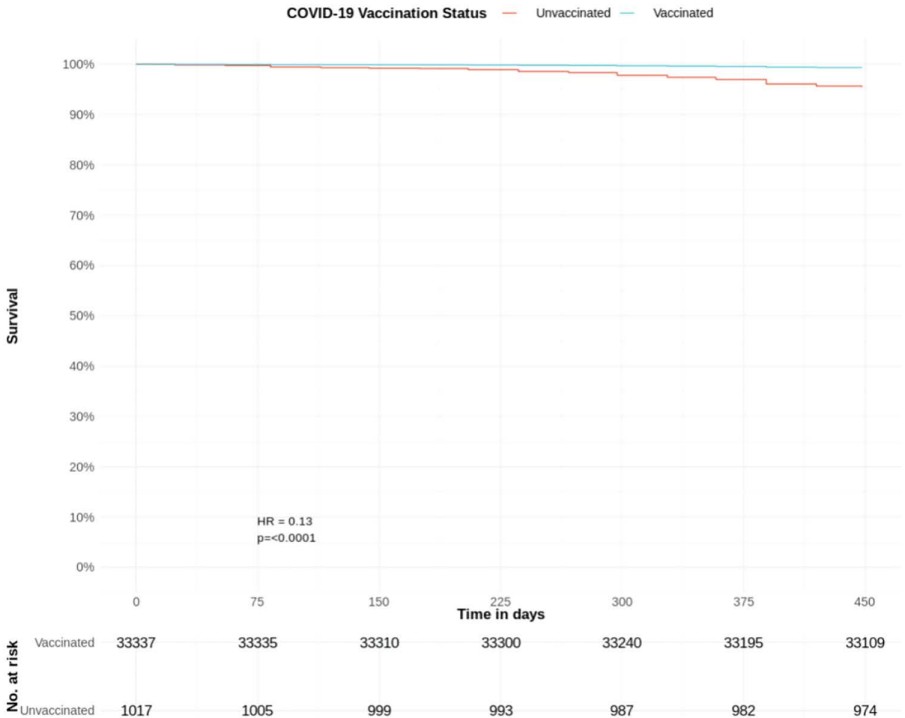

**Fig 8. COVID-19 mortality hazard in patients with CKD stage 3 adjusted for COVID-19 vaccination status: Survival curves from adjusted Cox proportional hazards model (Table 4).** T0 = December 8th, 2020.

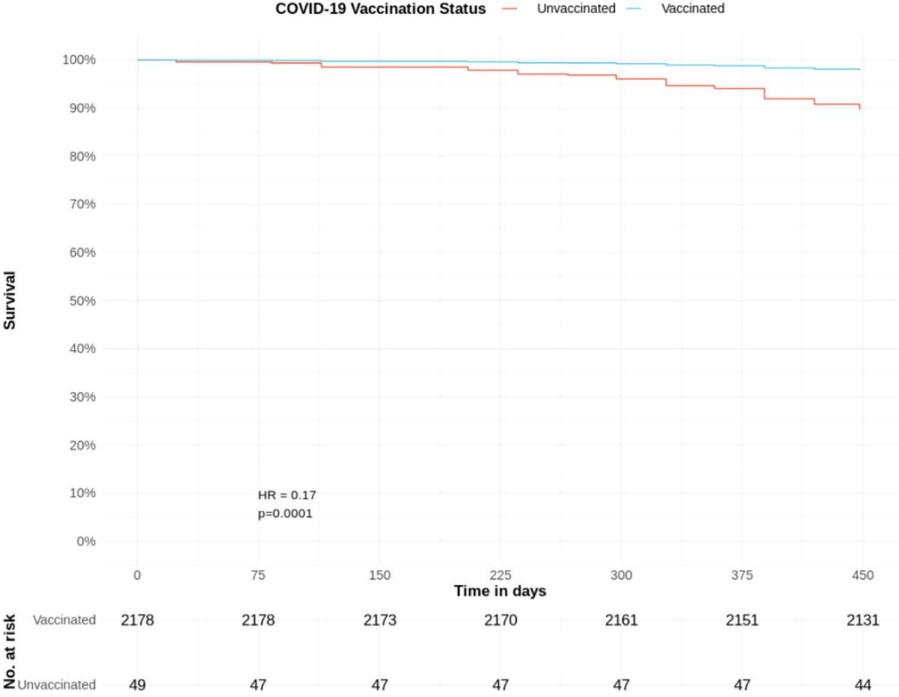

**Fig 9. COVID-19 mortality hazard in patients with CKD stage 4 and COVID-19 vaccination status: Survival curves from adjusted Cox proportional hazards model (Table 4).** T0 = December 8th, 2020.

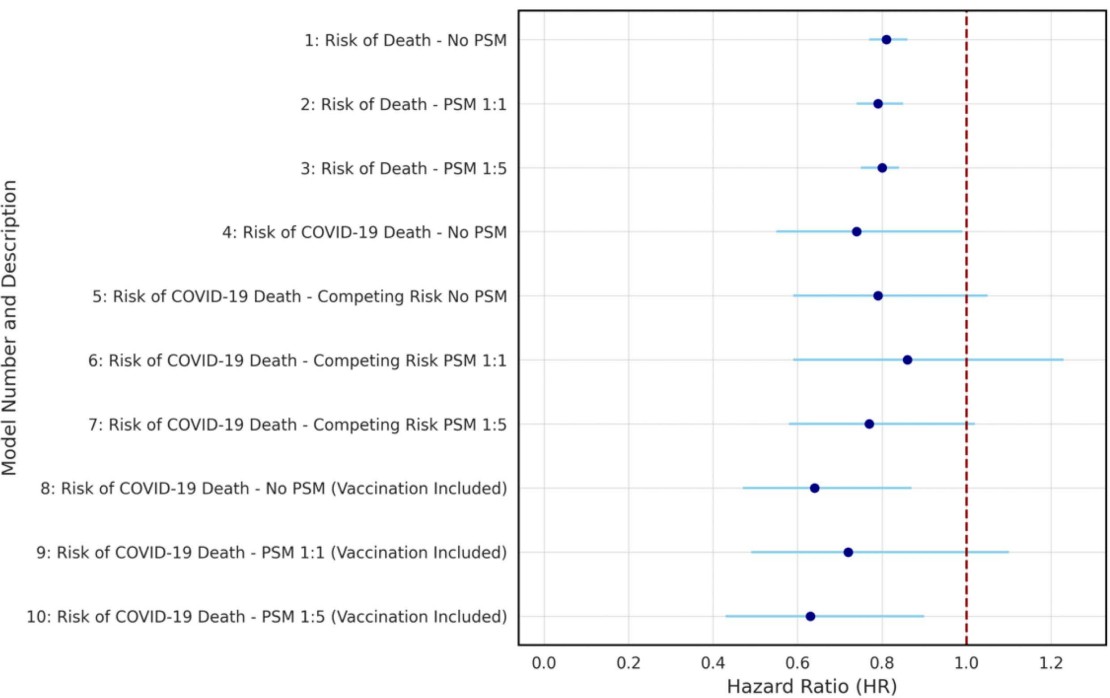

**Fig 10. Forest plot summary of adjusted hazard ratios from primary and sensitivity analyses. Key:** Rows 1-3 of each graph are adjusted hazard ratios and 95% CIs for primary analyses. Rows 4-10 show adjusted hazard ratios and 95% CIs for sensitivity analyses in Supporting information 3. Primary analyses describe a conditional hazard of death, whereas sensitivity analyses using CRR describe a marginal hazard of death.

These results reveal important information about mortality dynamics and competing risks in CKD patients. Existing evidence shows CKD stage 3 was associated with a lower risk of COVID-19 death than CKD stage 4 [22]. The CRR analyses may have failed to detect a significant result for CKD stage 3 due to the competing risk of non-COVID-19 deaths. The large cohort size of CKD stage 3 (n = 43550) introduces greater heterogeneity and opportunity for non-COVID-19 deaths.

Regarding CKD stage 4, CRR analyses reveal coding is associated with a lower adjusted hazard of a COVID-19 death, likely detectable due to higher COVID-19 mortality risk. It is important to acknowledge the smaller uncoded cohort of CKD stage 4 patients which may reflect inherent differences between the coded and uncoded groups. However, sensitivity analyses with CRR using PSM resulted in lower HRs indicating a greater impact of coding on COVID-19 mortality risk. These findings have implications for practice, policy and research.

Our findings emphasise the importance of clinical coding not only as a *step* in the diagnostic pathway but as an *intervention* to improve outcomes [4,9,21]. Coding provides automated care optimisation and prescribing alerts that are beneficial especially for older adults who are at greater risk of AKI, and adverse events due to polypharmacy [21].

Uncoded CKD is common [3,9,10] and barriers to coding CKD have been described [4]. The prevalence of undiagnosed CKD is higher in other high-income countries (stage 3 prevalence range: 61.6%−95.5%) [23] than in our analyses (stage 3 uncoded prevalence: 18.9%) in part because we excluded uncoded patients at baseline that were coded during the study period. In the UK, strategic initiatives to improve CKD coding practices in primary care [4,24,25] alongside collaborations between primary care and nephrologists may also contribute to lower rates of uncoded CKD in our primary care population [21,24,26–28]. These collaborations in England are supported by national auditing tools (CVD PREVENT

[12]) which allows clinicians to quickly estimate the number of patients at practice and regional level with uncoded CKD. Evidence from Japan, underscores the benefits of primary care-nephrology collaborations for managing patients with CKD stage 5 – collaborative practices were associated with a lower hazard of infection-related hospitalisation (HR 0.36; CIs 0.15–0.87) [29].

Despite these efforts, uncoded CKD remains a significant challenge. A recent study in England showed uncoded CKD prevalence at hospital discharge was 58.7% with only 6.7% receiving a diagnostic CKD code during hospitalisation [21]. This variability across settings illustrates the heterogeneity in coding practices and the universality of the problem requiring a collaborative population health approach [21].

## Strengths and limitations

Our research strengthens existing evidence [10] showing clinical coding is associated with a lower adjusted hazard of death for CKD stages 3–4 (whilst controlling for a greater selection of covariates) and reveals the association with a lower adjusted hazard of COVID-19 death. Analyses involved data from a large primary care dataset. We excluded patients with dementia and palliative clinical codes at study start recognising their likely association with increased mortality risks. Our findings contribute to the limited evidence base on the impact of mental health disease in CKD patients on risk of death and COVID-19 death. Sensitivity analyses robustly address covariate imbalances and the less explored scenarios of competing risks.

Limitations include comparing patients with uncoded CKD at study start who remained uncoded throughout the study period (never coded) with patients with coded CKD at study start. Diagnoses at study start do not capture quality or heterogeneity of care. Residual confounding in 1 PSM model for stage 4 CKD (risk of death; Supporting information 3) may overestimate specific CRR HRs. Primary analyses describe a conditional hazard of death; sensitivity analyses using CRR describe a marginal hazard of death – therefore the HRs in Cox PH models and CRR models are not directly comparable.

## Conclusions

Our retrospective cohort study suggests that clinical coding is an intervention associated with a reduced hazard of death and a possibly reduced hazard of COVID-19 death for patients with CKD stages 3 and 4 emphasising the importance of coding not only in clinical record keeping but also its potential in improving health outcomes.

## Supporting information

**Supporting information 1. Data dictionary.** Variable names and types.
(DOCX)

**Supporting information 2. Table 1 by coding status.** Clinical and demographic summary of cohort to show CKD stages 3 and 4 covariate prevalence stratified by coding status.
(DOCX)

**Supporting information 3. Sensitivity Analyses.**
(DOCX)

## Acknowledgments

Professor Tom Blakeman is supported by the National Institute for Health and Care Research (NIHR) Greater Manchester Patient Safety Research Collaboration (GM PSRC). The views expressed are those of the author(s) and not necessarily those of the GM PSRC, the NIHR or the Department of Health and Social Care. We acknowledge the use of artificial intelligence-assisted tools (ChatGPT; GPT-4o model) in troubleshooting R code, and reducing repetition to improve clarity

of the manuscript. The following prompt was used for the latter point: "Please help me improve the flow and clarity by highlighting areas of unnecessary repetition, duplication, or ambiguity". All intellectual content, interpretation, and conclusions are the sole responsibility of the authors.

## Author contributions

**Conceptualization:** Stuart Stewart, Philip A. Kalra, Smeeta Sinha.

**Data curation:** Stuart Stewart, George Tilston.

**Formal analysis:** Stuart Stewart.

**Funding acquisition:** Stuart Stewart.

**Investigation:** Stuart Stewart.

**Methodology:** Stuart Stewart, Philip A. Kalra, Evangelos Kontopantelis, Smeeta Sinha.

**Project administration:** Stuart Stewart, George Tilston.

**Resources:** Stuart Stewart.

**Software:** Stuart Stewart.

**Supervision:** Philip A. Kalra, Evangelos Kontopantelis, Tom Blakeman, Smeeta Sinha.

**Validation:** Stuart Stewart.

**Visualization:** Stuart Stewart.

**Writing – original draft:** Stuart Stewart, Philip A. Kalra, Evangelos Kontopantelis, Tom Blakeman, Smeeta Sinha.

**Writing – review & editing:** Stuart Stewart, Philip A. Kalra, Evangelos Kontopantelis, Tom Blakeman, Smeeta Sinha.

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
