## [Decision Letter · Decision Letter 0]

30 Jul 2025

PONE-D-25-26228Quantifying the impact of clinical coding in chronic kidney disease on risk of death and COVID-19 deathPLOS ONE

Dear Dr. Stuart Stewart,   

Thank you for submitting your manuscript to PLOS ONE. After careful consideration, we feel that it has merit but does not fully meet PLOS ONE’s publication criteria as it currently stands. Therefore, we invite you to submit a revised version of the manuscript that addresses the points raised during the review process.

 Please submit your revised manuscript by Sep 13 2025 11:59PM. If you will need more time than this to complete your revisions, please reply to this message or contact the journal office at plosone@plos.org . Please include the following items when submitting your revised manuscript:

We look forward to receiving your revised manuscript.

Kind regards,

Diego Moriconi

Academic Editor

PLOS ONE

Journal Requirements:

3. Thank you for stating the following in the Competing Interests section*:

“I have read the journal's policy and the authors of this manuscript have the following competing interests: SSt receives funding from Kidney Research UK; award number AHPF_001_20220705. PK received grant funding from CSL Vifor and Astellas, consulting fees from Astra Zeneca, Vifor, Unicyte and UCB, honoraria from Astra Zeneca, Pfizer, Pharmacosmos, Medice, GSK, Bayer and CSL Vifor. EK – None to declare. TB – No financial conflicts of interest; NHS England Think Kidneys Programme Board Member (2014–17); Royal College of General Practitioners’ AKI Clinical Champion (2017–20); NHSE Renal Services Transformation Programme Post‑AKI care Lead (2021–23); Specialist Committee Member for NICE AKI Quality Standard (QS76) (2022–23); Kidney Disease Improving Global Outcomes (KDIGO) AKI Guideline Work Group (2023‑To date). GT – none to declare. SSi received honoraria from AZ, Boehringer, Menarini, Novartis, GSK, CSL Vifor and Bayer.”

We note that one or more of the authors have an affiliation to the commercial funders of this research study

“GT is funded by the National Institute for Health and Care Research (NIHR) Manchester Biomedical Research Centre (BRC) (NIHR203308).”

“Dr Stuart Stewart receives doctoral research funding from Kidney Research UK (AHP_001_202207405). The funder of the study had no role in article design, analysis, or publication writing.”

Additional Editor Comments (if provided):

Dear Authors,

after careful evaluation and consideration of the reviewers' comments, we believe that your manuscript has potential but requires substantial revisions before it can be considered for publication. Therefore, we are inviting you to submit a major revision.

Please address all the reviewers' comments thoroughly in your revised manuscript and provide a detailed response to each point raised. This will help us and the reviewers assess how the concerns have been addressed.

We appreciate the value of your work and look forward to receiving your revised manuscript, along with your responses to the reviewers’ remarks.

Reviewers' comments:

Reviewer's Responses to Questions

**Comments to the Author**

1. Is the manuscript technically sound, and do the data support the conclusions?

Reviewer #1: Yes

Reviewer #2: Yes

2. Has the statistical analysis been performed appropriately and rigorously? 

Reviewer #1: Yes

Reviewer #2: No

3. Have the authors made all data underlying the findings in their manuscript fully available?

Reviewer #1: Yes

Reviewer #2: Yes

4. Is the manuscript presented in an intelligible fashion and written in standard English?

Reviewer #1: Yes

Reviewer #2: Yes

5. Review Comments to the Author

Reviewer #1: 1. Originality

This is an original study as it highlights that CKD coding is associated with a downstream alert signal for patients, potentially regardless of the medical specialty they are subsequently seen by. It emphasizes associated comorbidities and the possible impact on the overall clinical evaluation of these patients.

2. Importance

In a well-structured and educated healthcare system, CKD coding is a key element for the systematic monitoring of these patients and enables proper, structured management. It also facilitates early detection and highlights potential complications in their disease progression.

4. Results

Regardless of the future healthcare setting or possible instability in patient care, CKD coding remains a crucial element that benefits both the medical system and the patients.

Questions:

1. Has any analysis been conducted on the patient follow-up rate after CKD coding?

2. Is there any analysis of the financial impact resulting from CKD coding?

3. Has there been an age-specific analysis on CKD coding—particularly in the elderly population?

Reviewer #2: Thank you for giving me the opportunity to review the manuscript. This is a retrospective cohort study of patients with stage 3–5 chronic kidney disease (CKD). The authors reported that uncoded CKD was associated with a higher risk of all-cause and coronavirus disease 2019 (COVID-19) mortality. Although this is a thought-provoking study, several concerns need to be addressed.

Major comments

1) My greatest concern is statistical analysis. The aim of this study was to investigate the association between coding of CKD and all-cause and COVID-19 mortality among patients with stage 3–5 CKD. Therefore, I would suggest that the authors include all patients with stage 3–5 CKD in the main analysis. After conducting this main analysis, the authors need to perform the same analysis by subgroups of CKD (stages 3 and 4 CKD).

2) Given that the aim of this study was to investigate the association between coding of CKD and mortality, I would suggest that the authors compare the two groups in Table 1: patients with coded CKD and those with uncoded CKD.

3) Considering the study design and limitations of this study, I would suggest that the authors town the Conclusion section.

4) I would suggest that the authors clearly describe the definition of confirmed SARS-CoV-2 infection. Were the patients diagnosed with COVID-19 based on positive PCR or antigen test results?

5) I would suggest adding discussions regarding the high proportion of coded CKD and the role of primary care physicians in England. Surprisingly, 81.1% of the patients were coded as having CKD in this study. A previous study using databases from five countries (France, Germany, Italy, Japan, and the United States) reported consistently high proportions of patients with undiagnosed stage 3 CKD [Ref #1]. This discrepancy can, at least in part, be attributed to the quality of primary care physicians in England. A previous study reported that preventing hospitalization is a key role for primary care physicians, even during the COVID-19 pandemic [Ref #2]. Primary care physicians play an important role in the field of nephrology. A cohort study in Canada reported that patients with stage 1–4 CKD treated by primary care physicians had a lower risk of hospitalization for heart failure and hyperkalemia than those who were not treated by primary care physicians [Ref #3]. Regarding infections, primary care physician-nephrologist collaboration was associated with a lower risk of infection-related hospitalization among patients with stage 5 CKD [Ref #4]. The above discussions on the role of primary care physicians will enhance the importance of this study.

Ref #1 Tangri N, et al. Prevalence of undiagnosed stage 3 chronic kidney disease in France, Germany, Italy, Japan and the USA: results from the multinational observational REVEAL-CKD study. BMJ Open. 2023;13:e067386.

Ref #2 Aoki T, et al. Impact of primary care attributes on hospitalization during the COVID-19 pandemic: a nationwide prospective cohort study in Japan. Ann Fam Med 2023;21:27–32.

Ref #3 Wiebe N, et al. Potentially preventable hospitalization as a complication of CKD: a cohort study. Am J Kidney Dis 2014;64:230–238.

Ref #4 Murakami M et al. Association between primary care physician–nephrologist collaboration and clinical outcomes in patients with stage 5 chronic kidney disease: a JOINT‑KD cohort study. J Nephrol 2025;38:1385–1394.

6) I would suggest adding a supplementary figure showing the flowchart of the study participants because only 47,628 patients with stage 3–5 CKD out of 2.45 million patients were included in the analysis.

7) I would suggest adding the causes of CKD to Table 1. It is unclear whether diseases, such as diabetes and hypertension, are the causes of CKD or comorbidities.

Minor comments

8) I would suggest that the authors clarify why patients who died due to non-COVID-19 were excluded from the analysis (page 6). In the Cox proportional hazards model, the patients who experienced such events are censored. However, in the Fine–Gray model, such events are treated as competing risk events.

9) I would suggest adding a supplementary Figure showing the flowchart of the study participants because only 47,628 patients with stage 3–5 CKD out of 2.45 million patients were included in the analysis.

10) I would suggest that the authors clearly describe whether the hazard ratio is unadjusted or adjusted throughout the manuscript. The term “hazard ratio” is ambiguous for readers to understand the results of this study.

11) In relation to the above comment, I would suggest that Tables 2 and 3 list the adjusted hazard ratios, 95% confidence intervals, and p-values in that order.

12) I would suggest that the authors revise the Descriptive analyses by CKD stage section. As mentioned above, descriptions according to the CKD stage are not necessary because this is a subgroup of study participants. Moreover, these sentences appear redundant.

13) I would suggest combining Figures 3 with 4, and 5 with 6. A Kaplan–Meier curve with an enlarged y-axis is commonly incorporated within the area of the original Kaplan–Meier curve.

14) I would suggest that the author suggests clearly stating the p-value (page 15). when the p-value is 0.05, please include three decimal places (e.g., 0.048).

15) I would suggest that the authors recheck the reference list. For example, #20 of Reference needs to be revised as follows:

BMC Nephrol 2025;26:1–10. → BMC Nephrol 2025;26:39.

16) I would suggest that the authors correct grammatical errors throughout the manuscript.

6. PLOS authors have the option to publish the peer review history of their article (what does this mean? ). If published, this will include your full peer review and any attached files.

**Do you want your identity to be public for this peer review?** For information about this choice, including consent withdrawal, please see our Privacy Policy .

Reviewer #1: No

Reviewer #2: No

---

## [Author Response · Author response to Decision Letter 1]

11 Sep 2025

Please see our word document with response to peer-reviewers

---

## [Decision Letter · Decision Letter 1]

22 Sep 2025

Quantifying the impact of clinical coding in chronic kidney disease on risk of death and COVID-19 death

PONE-D-25-26228R1

Dear Dr. Stuart Stewart,

We’re pleased to inform you that your manuscript has been judged scientifically suitable for publication and will be formally accepted for publication once it meets all outstanding technical requirements.

Kind regards,

Diego Moriconi

Academic Editor

PLOS ONE

**Comments to the Author**

1. If the authors have adequately addressed your comments raised in a previous round of review and you feel that this manuscript is now acceptable for publication, you may indicate that here to bypass the “Comments to the Author” section, enter your conflict of interest statement in the “Confidential to Editor” section, and submit your "Accept" recommendation.

Reviewer #1: All comments have been addressed

Reviewer #2: (No Response)

2. Is the manuscript technically sound, and do the data support the conclusions?

Reviewer #1: Yes

Reviewer #2: Yes

3. Has the statistical analysis been performed appropriately and rigorously? 

Reviewer #1: Yes

Reviewer #2: Yes

4. Have the authors made all data underlying the findings in their manuscript fully available?

Reviewer #1: Yes

Reviewer #2: Yes

5. Is the manuscript presented in an intelligible fashion and written in standard English?

Reviewer #1: Yes

Reviewer #2: Yes

6. Review Comments to the Author

Reviewer #1: Good day,

I notice differences between the original article and the revised version, with a significant improvement in the latter.

I believe it provides a comprehensive overview, analyzing the issues in detail and offering a broad perspective on the impact of COVID-19 in CKD.

I consider it suitable for publication. Congratulations to the authors for their work.

Reviewer #2: Thank you for revising the manuscript. It has significantly improved from the original submission, and the authors deserve commendations for their continued efforts. However, I would suggest that the authors revisit the conclusion section. Given the limitations of a cohort study, I would strongly suggest prefacing the sentences with “Our (retrospective cohort) study suggests that.” For instance, the following concluding sentences may be appropriate for this study:

Conclusion section in the Abstract

Our retrospective cohort study suggests that clinical coding is associated with a lower risk of all-cause and COVID-19 death in patients with CKD stages 3 and 4 and should be considered a key element in the organization and delivery of care for people with CKD.

Conclusion section

Our retrospective cohort study suggests that clinical coding is associated with a reduced risk of all-cause and COVID-19 death in patients with CKD stages 3 and 4, emphasizing the importance of coding not only in clinical record keeping but also in its potential to improve health outcomes.

7. PLOS authors have the option to publish the peer review history of their article (what does this mean? ). If published, this will include your full peer review and any attached files.

**Do you want your identity to be public for this peer review?** For information about this choice, including consent withdrawal, please see our Privacy Policy .

Reviewer #1: No

Reviewer #2: No

---

## [Editor Report · Acceptance letter]

PONE-D-25-26228R1

PLOS ONE

Dear Dr. Stewart,

I'm pleased to inform you that your manuscript has been deemed suitable for publication in PLOS ONE. Congratulations! Your manuscript is now being handed over to our production team.

Kind regards,

on behalf of

Dr. Diego Moriconi

Academic Editor

PLOS ONE